

# Constructing age-structured matrix population models for all fishes

Masami Fujiwara

Department of Ecology and Conservation Biology, Texas A&M University, College Station, TX,
United States of America

Corresponding author
Masami Fujiwara, fujiwara@tamu.edu

## ABSTRACT

Matrix population models are essential tools in conservation biology, offering key metrics to guide species management and conservation planning. However, the development of these models is often limited by insufficient life history data, particularly for non-charismatic species. This study addresses this gap by using life history data from FishBase and the FishLife R package, complemented by size-dependent natural mortality estimates, to parameterize age-structured matrix population models applicable to most fish species. The method was applied to 30 fish species common around oil and gas platforms in the Northern Gulf of Mexico, generating seven key metrics: damping ratio, resilience, generation time, stable age distribution, reproductive value, sensitivity matrix, and elasticity matrix. The damping ratio reflects how quickly a population returns to a stable age distribution after a disturbance, while resilience indicates the speed of recovery from perturbations. Generation time captures the average age of reproduction, and the stable age distribution represents the long-term proportion of individuals in each age class. Reproductive value quantifies future reproductive potential by age class. The sensitivity matrix highlights the age-class transitions most affecting population growth, and the elasticity matrix shows the proportional influence of these factors on population growth. The results demonstrate that robust population models can be constructed with limited species-specific data and reveal notable differences in population dynamics among species. For example, species with longer generation times, like the greater barracuda (*Sphyraena guachancho*), have lower damping ratios, indicating prolonged transient dynamics. In contrast, species such as the round scad (*Decapterus punctatus*) exhibit shorter generation times and higher damping ratios, suggesting faster returns to equilibrium. These findings underscore the importance of life history variability in shaping conservation strategies. Additionally, metrics like stable age distributions and reproductive values provide insight into population structure and individual contributions to future populations, while sensitivity and elasticity matrices inform management interventions such as size limits in fisheries. By integrating extensive databases and predictive tools, this study offers a scalable approach for developing matrix population models across diverse fish species. This methodology enhances our understanding of fish population dynamics, particularly for data-deficient species, and supports more informed conservation efforts. It also promotes ecosystem-based management by enabling species comparisons through standardized metrics, contributing to the sustainability of marine ecosystems.

## INTRODUCTION

Matrix population models are simple and practical tools commonly used in species conservation (*e.g.*, *Crouse, Crowder & Caswell, 1987*; *Fujiwara & Caswell, 2001*). These models allow for the calculation of various metrics essential for conservation planning. For example, the damping ratio measures how quickly short-term fluctuations in population density (transient dynamics) dissipate (*Caswell, 2001*), and generation time indicates the time scale of population dynamics (*Bienvenu & Legendre, 2015*). Transient dynamics can potentially mislead the status of a population (*Wiedenmann, Fujiwara & Mangel, 2009*), and generation time is one of the key parameters used in the IUCN Red List Criteria (*IUCN, 2012*). Stable age distribution and reproductive value indicate the expected proportion of individuals among groups and the relative importance of individuals in different groups toward future abundance, respectively. Any deviation from stable age distribution can diagnose past population changes (*e.g.*, *Williams et al., 2011*), and reproductive value can help allocate conservation efforts (*e.g.*, *Howe & Knopf, 1991*).

Although these metrics are useful, parameterizing matrix population models often requires extensive data sets. Common data types for parameterizing these models are life table data (*Fujiwara & Diaz-Lopez, 2017*) and capture-recapture data (*Nichols et al., 1992*). However, obtaining such data is typically limited to charismatic species of conservation interest or economic importance. Furthermore, building matrix population models requires estimates of survival and reproductive rates covering almost the entire life history of organisms, which are rare.

The lack of data is a general issue in conservation biology, not just for constructing matrix population models. For example, many species on the IUCN Red List are classified as data deficient, with unknown status (*IUCN, 2024*). To address data deficiencies, current efforts focus on compiling data from various sources and building species databases. While much progress has been made for various taxa (*e.g.*, *AmphibiaWeb, 2024*; *Avibase, 2024*; *BirdLife International, 2024*; *Froese & Pauly, 2024*; *Kew Gardens, 2024*; *PanTHERA, 2024*; *Reptile Database, 2024*; *The Plant List, 2024*; *Wilson & Reeder, 2024*), most databases still primarily include taxonomic or spatial distribution information. Moreover, the accuracy and completeness of these databases may be variable, necessitating further validation. Nonetheless, these limitations should not prevent the development of methods to effectively utilize the available data within these resources.

For fish species, FishBase (*Froese & Pauly, 2024*) is a comprehensive database that includes life history data for many species. Furthermore, *Thorson et al. (2017)* and *Thorson (2019)* developed algorithms to fill in missing life history data in FishBase using taxonomic relatedness. This algorithm is available as the FishLife R package (*Thorson, 2024*). The objective of this study is to use life history data from the FishLife package, supplemented

with additional data from FishBase and size-dependent natural mortality estimates (*Lorenzen, 1996*), to parameterize matrix population models. This method can be applied to almost all known fish species.

Here, the method was applied to fish species inhabiting oil and gas platforms in the Northern Gulf of Mexico. These offshore structures have become vital habitats for a variety of fish species and are key locations for recreational fisheries. With over 10,000 platforms constructed in the Gulf, they have also become integral to recreational fishing communities. However, as many of these platforms face decommissioning, understanding the potential impact on associated fish communities is an emerging area of interest (*Fujiwara, Beyea & Putman, 2024*). Notably, many of these species are not the primary targets of commercial fisheries, leading to insufficient data for assessing the implications of population dynamics. Therefore, developing matrix population models for data-deficient species is crucial. To illustrate the utility of the proposed method in constructing these models, the method was applied to 30 fish species found around oil platforms in the Northern Gulf of Mexico. With each estimated population matrix, seven key metrics were calculated. These are damping ratio, resilience, generation time, stable age distribution, reproductive value, sensitivity matrix, and elasticity matrix:

1. **Damping ratio**: This metric indicates how rapidly a population moves toward its stable age distribution. It is derived as the ratio of the dominant eigenvalue and the second largest eigenvalue from the projection matrix.

2. **Resilience:** In the context of matrix population models, resilience describes how quickly a population recovers to its equilibrium after being disturbed. This is often represented by the negative value of the dominant eigenvalue in a linearized density-dependent matrix.

3. **Generation time:** Generation time can have several definitions, but in the present analysis, it refers to the mean age at which individuals reproduce. It is a measure of the time it takes for a population to replace itself, often calculated by considering the weighted mean age of mothers at childbirth.

4. **Stable age distribution:** This term describes the consistent proportion of individuals in each age class that a population achieves over time, provided it grows at a steady rate. Once this distribution is reached, the relative abundance of each age group remains stable across generations.

5. **Reproductive value:** Reproductive value assesses the future reproductive contributions of individuals from a particular age class to population growth. It integrates both survival and fertility rates to estimate the expected reproductive output of individuals throughout their remaining life.

6. **Sensitivity matrix:** The sensitivity matrix identifies how variations in each element of the population projection matrix influence the population's growth rate ($\lambda$). It helps to determine which transition has the most impact on population dynamics.

7. **Elasticity matrix:** The elasticity matrix evaluates the proportional effect of each matrix element on the population growth rate ($\lambda$). Unlike the sensitivity matrix, it adjusts for the relative magnitudes of the matrix elements, offering a size-independent measure of their influence.

**Table 1  List of data used in constructing the population matrix.** Symbols are those used in the source database/paper (the same symbol is used to represent different data by different sources).

| Parameters | Description | Symbol | Source |
|---|---|---|---|
| Length-Mass | Allometric relationships for length-mass relationships | $a, aTL, b$ | FishBase, "length_weight" data |
| Length-Length | Total length to standard/fork length conversion | $a, b$ | FishBase, "length_length" data |
| Mortality | Coefficients for mass-specific instantaneous mortality rate | $M_u, b$ | Lorenzen (1996) |
| Maximum age | The maximum age | $tmax$ | FishLife Package (Thorson, 2019) |
| Age at maturity | The age at maturity | $tm$ | FishLife Package (Thorson, 2019) |
| Von Bertalanffy growth | The von Bertalanffy growth parameters | $Loo, K$ | FishLife Package (Thorson, 2019) |
| Length at maturity | The length at maturity | $Lm$ | FishLife Package (Thorson, 2019) |
| Density-dependent parameter | The maximum annual spawner biomass per spawner biomass in excess of replacement | $ln\_MASPS$ | FishLife Package (Thorson, 2019) |
| Covariance matrix | The covariance matrix for all of the parameters in the FishLife package | $cov\_pred$ | FishLife Package (Thorson, 2019) |

More detailed descriptions of these metrics can be found in *Caswell (2001)*.

## MATERIALS & METHODS

### Data

To construct an age-structured matrix population model, data from three sources were combined. Here, the term "data" refers to the statistics available from databases and publications rather than the original raw data. The first source was FishBase (*Froese & Pauly, 2024*), which is an online database of fishes. It includes life history and other data for 35,600 species. The amount of data varies substantially among species and depends on the number of researchers who contribute to the database. The data from FishBase were directly downloaded using the rfishbase package (v 2.0; *Boettiger, Lang & Wainwright, 2012*). The second source was the FishLife R package (*Thorson, 2024*). The package takes the life history data from FishBase and predicts missing life history parameters for other species based on their taxonomic relatedness (*Thorson et al., 2017*). A newer version of the FishLife package also uses data from the database of stock-recruitment relationships developed by *Myers, Bridson & Barrowman (1995)* to predict density-dependent parameters (*Thorson, 2019*). Finally, the third source of data was the general mass-dependent mortality of fish derived by *Lorenzen (1996)*. This data was used instead of the mortality estimates from FishBase, which often only provides mortality estimates independent of size or age (typically a point estimate for adults). Table 1 summarizes the life history data used in constructing the population matrix.

### Construction of age-structured matrix population models

The data were assembled into age-structured matrix population models as follows. More details for each step are described later in this section.

1. The size of the population matrix was determined based on the maximum age.
2. The age starting with 1 was converted into length using the von Bertalanffy growth equation.
3. The length was converted into mass using length-mass allometric relationships.

4. The mass was converted into the instantaneous mortality rate.
5. The instantaneous mortality rate was converted into the age-specific finite survival rate, which was entered into the subdiagonal elements of the age-strcutured matrix population model (see further details below).
6. For ages greater than the age of maturation, the fertility rate was calculated assuming it is proportional to the mass of individuals for the age and scaled such that the asymptotic growth rate is 1. (Density-Independent Matrix Population Model)
7. The density-dependent parameters were calculated from the maximum annual spawner biomass per spawner biomass in excess of replacement, age of maturity, age-specific individual mass, and age-specific survival rate. (see Density-Dependent Matrix Population Models)

## Determining the size of the population matrix

The size of a population matrix was determined by the maximum age ($tmax$), obtained from the FishLife package, rounded up to the nearest integer. Hereafter, this integer value is denoted as $t_{max}$. This assumes that the time unit of the model is one year. Therefore, only cases in which the maximum age is greater than 1 were considered. In other words, annual populations are not addressed in this paper because they lead to unstructured population models.

## Age to length conversion

Age $t$ was converted into length $L(t)$ using the von-Bertalanffy equation:

$$L(t) = L_\infty \left(1 - e^{(-\kappa(t-t_0))}\right) \tag{1}$$

where $L_\infty$ is the asymptotic length, $\kappa$ is the parameter determining how fast individuals grow in size, and $t_0$ is an extrapolated age at which the length would have been 0. $L_\infty$ and $\kappa$ were obtained from the FishLife package. $t_0$ was calculated by solving Eq. (1) for $t_0$ and substituting the age of maturity and the length of maturity (both also obtained from the FishLife Package) along with $L_\infty$ and $\kappa$. When the obtained value for $t_0$ was greater than $-0.1$, $-0.1$ was used instead; it is the default value used in *Thorson (2019)*. The unit of length in the package was in cm, and it assumes the total length (rather than standard length or fork length).

## Length to mass conversion

The age-specific mass $W(t)$ was used in determining the instantaneous mortality rate and was converted from the length $L(t)$ of individuals using the following allometric relationship:

$$W = aL^b \tag{2}$$

where $a$ and $b$ are the allometric coefficients obtained from FishBase. Note that Eq. (2) omits the age to reduce clutter, but both length and mass are a function of age. Here, the unit of mass is in grams.

The length in the length-mass relationship may be measured in terms of total length, fork length, or standard length. If the total length was used, the allometric parameters were

directly used in Eq. (2). When fork length or standard length was used, the total length $L(t)$ was converted into the corresponding length measure using the length-length relationship obtained from FishBase and substituted into Eq. (2). The length-length relationship is expressed as a linear equation. For example, the total length $L_T$ is converted into standard length $L_S$ as:

$$L_S = \alpha + \beta L_T \tag{3}$$

where $\alpha$ and $\beta$ are the intercept and slope coefficients, respectively, for the length-length relationships. Note that the original database uses symbols $a$ and $b$, but they were changed into $\alpha$ and $\beta$ to avoid using the same symbols for different quantities in this paper. The total length can be converted into the fork length $L_F$ in the same manner. Then, $L_S$ or $L_F$ was substituted into Eq. (2) instead of $L_T$ when the allometric relationship was for standard length or fork length, respectively.

## Natural Mortality

The natural mortality was obtained by using the Lorenzen mass-dependent mortality rate:

$$m = -\mu W^\upsilon \tag{4}$$

where $m$ is the instantaneous mortality rate. The two coefficients $\mu$ and $\upsilon$ are those obtained from *Lorenzen (1996)*. The estimated values are $\mu = 3.00$ and $\upsilon = -0.288$. In the original paper, these two coefficients were denoted by $M_u$ and $b$, respectively, but to reduce confusion with other parameters, the symbols were changed in this paper. The unit of the instantaneous mortality rate is per year (year$^{-1}$).

## Age-specific finite survival rate

The age-specific natural mortality rate $m$ was converted to the age-specific finite survival rate $s_t$ as follows:

$$s_t = \frac{exp\left(\int_1^{t+1} m(\tau)\,d\tau\right)}{exp\left(\int_1^t m(\tau)\,d\tau\right)}. \tag{5}$$

The finite survival rate is the proportion of individuals that survive from age $t$ to $t+1$, and it is calculated for $t = 1, \ldots, t_{max}-1$, where $t_{max}$ is the size of the population matrix. The finite survival rate was entered into the $<t+1, t>$ elements of the population matrix.

## Fertility rate

The fertility rate for individuals of age $t$ ($F_t$) was assigned to a population matrix based on the following assumptions:

1. Individuals start contributing to reproduction once they reach the age of maturity.
2. The first age class is for age 1.
3. The fertility rate for age $t$ is proportional to the mass at age $t$.
4. The finite population growth rate ($\lambda$) is 1.

Therefore, the $<1,t>$ element of the population matrix is given by $F_t = cW(t)$ for $t$ greater than or equal to the age of maturity ($t_{mat}$). Here, $c$ is a proportionality constant,

and the same value is applied to all age-classes of the same population. To obtain the age of maturity ($t_{mat}$), the value denoted by tm in the FishLife package was rounded up to the nearest integer. Finally, the value for $c$ was numerically searched such that the dominant eigenvalue ($\lambda$) of the population matrix was 1.

Finally, 0 was entered into empty elements of the population matrix to complete the density-independent population matrix.

## Density-dependent population matrix

The density-dependent population matrix was developed by modifying the density-independent population matrix based on the following assumptions:

1. All density-dependent processes take place between ages 0 and 1.
2. Density dependence operates through the total biomass.
3. The age of maturity is the age of recruitment.
4. Density dependence takes the form of compensatory density dependence,
5. The total number of mature fish at the equilibrium point is 10,000 individuals per unit area (an arbitrary value, as the habitat area for a population is often unknown).

Therefore, the density-dependent fertility rate $F_{t,x}$ for age $t$ at time $x$ is given as:

$$F_{t,x} = \frac{D_a W_t}{1 + D_b B_x} \qquad (6)$$

where $D_a$ and $D_b$ are density-dependent constants and

$$B_x = \sum_{t=t_{mat}}^{t_{max}} n_{t,x} W_t \qquad (7)$$

where $n_{t,x}$ is the density of individuals of age $t$ at time $x$.

The coefficient $D_a$ was determined from the maximum annual spawner biomass per spawner biomass in excess of replacement (MASPS) in the FishLife package as follows:

$$D_a = \frac{p}{M_{spawner} l_{mat}} \qquad (8)$$

where $p$ is the maximum lifetime spawner biomass per spawner biomass (*Myers, 2001*) and given as,

$$p = 1 + \frac{\gamma}{1 - \overline{S}} \qquad (9)$$

where $\overline{S}$ is the mean survival rate after maturation and $\gamma$ is the MASPS (see Eq. (7) in *Thorson, 2019*). The survivorship from age 1 to maturation is denoted by $l_{mat}$ and is calculated as $\prod_{t=1}^{t_{mat}-1} s_t$. The survivorship is included in the denominator of Eq. (8) because the MASPS calculation assumes that density dependence occurs somewhere between reproduction and recruitment, whereas the model in this paper assumes that the density dependence occurs somewhere between reproduction and age 1. Therefore, the age-structured population matrix in the current study includes survival from age 1 to the age of maturity separately. The spawner biomass per recruit is denoted by $M_{spawner}$, and it is calculated as $\sum_{t=t_{mat}}^{t_{max}} \left( \frac{l_t}{l_{mat}} W_t \right)$ where $l_t$ is the survivorship from age 1 to age $t$. Finally, the

**Table 2 Metrics calculated from population matrices.** More details of descriptions can be found in *Caswell (2001)*.

| Metrics | Symbol | Description |
|---|---|---|
| | Density-independent population matrix | |
| Stable age distribution | $w$ | Expected proportion of individuals among age classes |
| Reproductive value | $v$ | Expected relative contribution of individuals in corresponding age class toward future population growth and abundance |
| Damping ratio | $\rho$ | A measure of how quickly a short-term fluctuation attenuates before assuming the asymptotic change in abundance |
| Sensitivity matrix | $S$ | How sensitive the asymptotic population growth rate is to changes in elements of a population matrix |
| Elasticity matrix | $E$ | How sensitive the asymptotic population growth rate is to proportional changes in elements of a population matrix |
| Generation time | $G$ | The mean age of parents of a given cohort |
| | Density-dependent population matrix | |
| Resilience | $R$ | A measure of how quickly a population returns to an equilibrium point after a perturbation |

value of $D_b$ is determined numerically, by setting $\sum_{t=t_{mat}}^{t_{max}} n_t = 10,000$, which is arbitrarily determined as it depends on the habitat in which a population lives (*i.e.,* rarely known parameter).

## Matrix population model analysis

Once the density-independent and density-dependent population matrices were obtained, various analyses were performed. Table 2 shows the seven quantities calculated in this study.

An age-structured population matrix (often denoted by **A**) is called a Leslie matrix (*Leslie, 1977*), and it takes the following form:

$$A = \begin{bmatrix} 0 & 0 & \cdots & F_t & F_{t_{max}} \\ s_1 & 0 & 0 & 0 & 0 \\ 0 & s_2 & 0 & 0 & 0 \\ 0 & 0 & \ddots & 0 & 0 \\ 0 & 0 & 0 & s_t & 0 \end{bmatrix} \tag{10}$$

where the subdiaconal elements are the age-specific survival rates, and the first row is for the fertility rates (the product of fecundity and survival of offspring until age 1; see *Kendall et al., 2019*). The fertility rate is assumed to be 0 before individuals mature. In other words, the <1,t>element is 0 if $t < t_{mat}$.

Once the density-independent population matrix **A** is obtained, the asymptotic population growth rate $\lambda_1$ is given by the dominant eigenvalue of the matrix. The associated right eigenvector gives the stable age distribution ($w$), which is often scaled to sum to 1. The associated left eigenvector gives the reproductive value ($v$), which is scaled in various ways.

The sensitivity matrix ($S$) is given by the product of the reproductive value ($v$) and stable age distribution ($w$) as

$$S = \frac{vw^T}{v^T w} \tag{11}$$

where both $v$ and $w$ are assumed to be column vectors. The $<i,j>$element ($e_{ij}$) of elasticity matrix ($E$) is given by $e_{ij} = \frac{a_{ij}\varsigma_{ij}}{\lambda_1}$ where $a_{ij}$ is the $<i,j>$element of the population matrix and $\varsigma_{ij}$ is the $<i,j>$element of the sensitivity matrix.

The damping ratio measures how quickly transient dynamics dissipate and the population returns to asymptotic growth/decline after a perturbation (*Caswell, 2001*). It is given by the ratio of the dominant eigenvalue and the second eigenvalue as

$$\rho = \frac{\lambda_1}{|\lambda_2|}. \tag{12}$$

Generation time has several different definitions. Here, it is defined as the expected mean age of the parents of newborn individuals in a cohort. It is calculated as

$$G = \frac{\lambda_1 v^T w}{v^T F w} \tag{13}$$

where $F$ is a matrix consisting of only the fertility rates (*Bienvenu & Legendre, 2015*).

Finally, the resilience $R$ is given by the dominant eigenvalue of the linearized population matrix $\lambda_J$ (*Caswell & Neubert, 2005*) as

$$R = -\ln\left(|\lambda_J|\right). \tag{14}$$

The linearized population matrix $J$ is given as

$$J = A|_{n^*} + \left[ \frac{\partial A}{\partial n_1}\bigg|_{n^*} n^*, \frac{\partial A}{\partial n_2}\bigg|_{n^*} n^*, \dots \frac{\partial A}{\partial n_k}\bigg|_{n^*} n^* \right] \tag{15}$$

where $n^*$ is a population density vector at the equilibrium and $n_i$ is the density of age class $i$.

## Quantification of uncertainties and computation

To quantify the uncertainties associated with the metrics, the data obtained from the FishLife package were simulated using the mean and variance from the package, assuming a multivariate normal distribution. Under each simulation, age-structured matrix population models (both density-independent and -independent models) were constructed, and all metrics were calculated. For the analysis presented in the paper, the data for each species were simulated 1,000 times.

All analyses were performed using R Statistical Software (v4.3.1; *R Core Team, 2024*) with the tidyverse package (v2.0.0; *Wickham et al., 2019*) and the MASS package (v 7.3; *Venables & Ripley, 2002*). The code used for the case study can be found in *Fujiwara (2024)*.

## Writing

After the draft of the manuscript was written, it was uploaded to ChatGPT-4o (*Open AI, 2024*) with memory off. The text was revised using the following prompt:

"I am writing a scientific paper to be published in a peer-reviewed journal in the field of ecology. The following is the draft of the manuscript. Please revise it for grammatical errors. Please also make the tense of verbs consistent with scientific writing practices. Keep the common names of species in singular form and lowercase without any article".

The manuscript was revised one section/subsection at a time. Then, the abstract was written using the following prompt:

"I am writing a scientific paper to be published in a peer-reviewed journal in the field of ecology. The following is the draft of the manuscript. Please write an abstract of the paper in approximately 300 words".

After all these processes, the manuscript was further edited manually for accuracy.

## RESULTS

To demonstrate the analysis, matrix population models for 30 species of fish found around oil and gas platforms in the northern Gulf of Mexico (Table 3; *Fujiwara, Beyea & Putman, 2024*) were constructed, and associated analyses were performed to obtain the seven metrics. Some of these species are targeted by recreational or commercial fishing, but others are not economically important. Consequently, complete data to build matrix population models for most of the species are missing.

The damping ratio, generation time, and resilience of the equilibrium point are shown in Figs. 1, 2 and 3, respectively. In each figure, species are ordered based on their median values, which are indicated by the vertical bars. The stable-age distribution and reproductive value of selected species are shown in Figs. 4 and 5, respectively. For these figures, nine species were arbitrarily selected. Finally, the sensitivity and elasticity matrices for the same nine species are shown in Figs. 6 and 7, respectively.

The damping ratios and generation times for the species studied provide insight into the dynamics of population responses to perturbations. A clear relationship exists between these two metrics, where species with longer generation times tend to have lower damping ratios (Figs. 1 and 2). For example, greater barracuda (*Sphyraena guachancho*) displayed the lowest median damping ratio and the longest median generation time, whereas round scad (*Decapterus punctatus*) exhibited the highest median damping ratio and shortest generation time. The resilience of equilibrium points for different species (Fig. 3) shows that snapper species generally have smaller median resilience compared to other species. This indicates that these species may experience a slow recovery after population declines. Stable age distributions and reproductive values (Figs. 4 and 5) reveal that cobia (*Rachycentron canadum*) and greater amberjack (*Seriola dumerili*) maintain a higher proportion of older individuals, whereas snapper species tend to lose individuals at younger ages but have a longer lifespan. The sensitivity and elasticity matrices (Figs. 6 and 7) highlight how changes in different population matrix elements affect population growth rates. For cobia and greater amberjack, the elasticity matrix suggests that size limits may not be the most effective management approach, while for species like lookdown and gulf menhaden, targeting younger individuals may yield more effective population management outcomes.

**Table 3** **Species of fish found around the oil and gas platforms in the northern Gulf of Mexico.** The table shows common names and scientific names.

| Common name | Scientific name |
| --- | --- |
| African Pompano | *Alectis ciliaris* |
| Sheepshead | *Archosargus probatocephalus* |
| Gray Triggerfish | *Balistes capriscus* |
| Spanish Hogfish | *Bodianus rufus* |
| Gulf Menhaden | *Brevoortia patronus* |
| Blue Runner | *Caranx crysos* |
| Crevalle Jack | *Caranx hippos* |
| Horse-eye Jack | *Caranx latus* |
| Black Jack | *Caranx lugubris* |
| Bar Jack | *Caranx ruber* |
| Atlantic Spadefish | *Chaetodipterus faber* |
| Atlantic Bumper | *Chloroscombrus chrysurus* |
| Round Scad | *Decapterus punctatus* |
| Rainbow Runner | *Elagatis bipinnulata* |
| Bermuda Chub | *Kyphosus sectatrix* |
| Red Snapper | *Lutjanus campechanus* |
| Gray Snapper | *Lutjanus griseus* |
| Dog Snapper | *Lutjanus jocu* |
| Leatherjack | *Oligoplites saurus* |
| Bluefish | *Pomatomus saltatrix* |
| Cobia | *Rachycentron canadum* |
| Vermilion Snapper | *Rhomboplites aurorubens* |
| Red Drum | *Sciaenops ocellatus* |
| King Mackerel | *Scomberomorus cavalla* |
| Atlantic Moonfish | *Selene setapinnis* |
| Lookdown | *Selene vomer* |
| Greater Amberjack | *Seriola dumerili* |
| Almaco Jack | *Seriola rivoliana* |
| Great Barracuda | *Sphyraena barracuda* |
| Guaguanche Barracuda | *Sphyraena guachancho* |

## DISCUSSION

This study introduces a method for constructing age-structured matrix population models for fish. It takes advantage of the algorithm introduced by *Thorson et al. (2017)* to fill in missing data based on taxonomic relatedness to other species. This allows the construction of matrix population models for species with limited data. Consequently, models can be constructed for almost any known fish species. The new method is particularly useful for species with missing data. For example, species such as lookdown (*Selene vomer*) and blue runner (*Caranx crysos*) in this study are common in reefs, but their demographic data are incomplete. Matrix population models for these species can aid in their management.

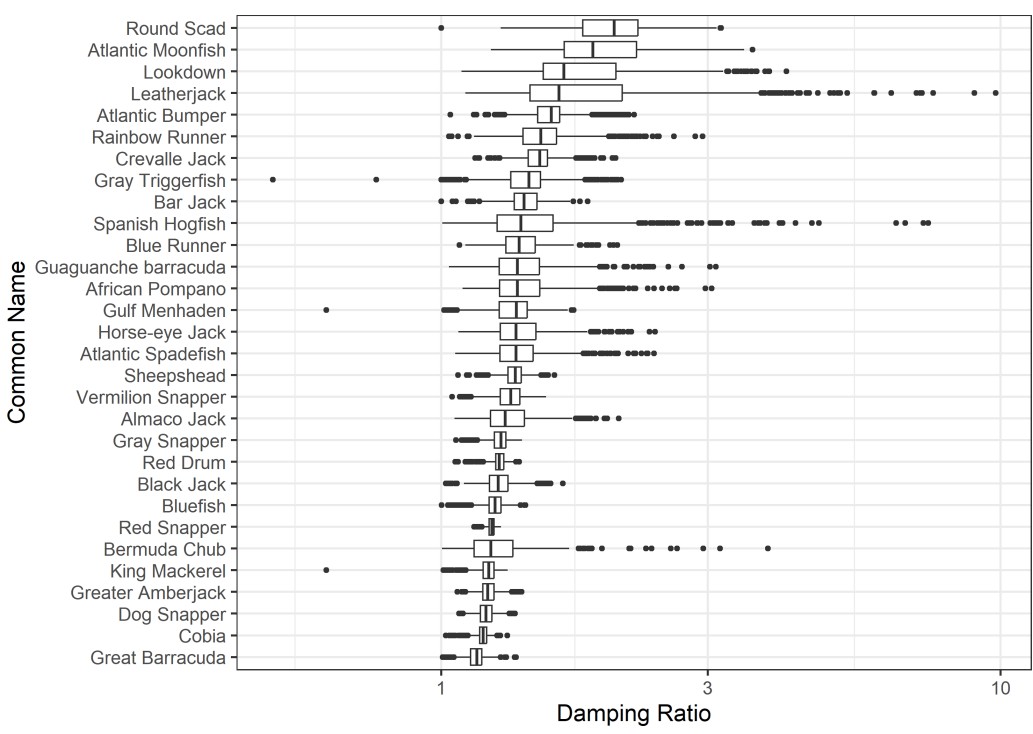

**Figure 1** **The damping ratio of 30 species of fish found at oil and gas platforms in the Northern Gulf of Mexico.** A box plot shows the median (vertical bar), upper and lower quartile (box), 1. 5 x Inter Quartile Range (IQR) below or above quartile range (whiskers), and outliers (dots).

The method also uses the model for size-dependent natural mortality of fish (*Lorenzen, 1996*). Therefore, the constructed matrix population models do not include fishing mortality. Consequently, the results obtained from the analysis are expected values under no-fishing activities. Lorenzen's model is one of several general mortality models for marine organisms. Other estimators include the Peterson and Wroblewski model (*Peterson & Wroblewski, 1984*) and the Sekharan model (*Sekharan, 1974*). The Lorenzen model was used in this study because it is a general mortality model for marine and freshwater fishes. On the other hand, the Peterson and Wroblewski model is for marine pelagic species, including fish and invertebrates, and the Sekharan model is specific to tropical fish species (see *Sierra Castillo & Fujiwara, 2021*; *Sierra Castillo, Pawluk & Fujiwara, 2020*). The mortality estimates from the FishLife package were not used in this study because the package provides a single estimate for a given species (*i.e.,* not size- or age-dependent). Furthermore, the estimate is based on the value in FishBase, which may or may not include fishing mortality, depending on the researchers reporting the data.

Fisheries managers are increasingly aware of the importance of non-focal species such as prey and predators for the sustainability of a focal species and advocate for ecosystem-based management (*Pikitch et al., 2004*). However, data are often missing for many of the non-focal species. Once matrix population models are built for those species, metrics, such as those calculated in this study (*i.e.,* damping ratio, generation time, resilience, age
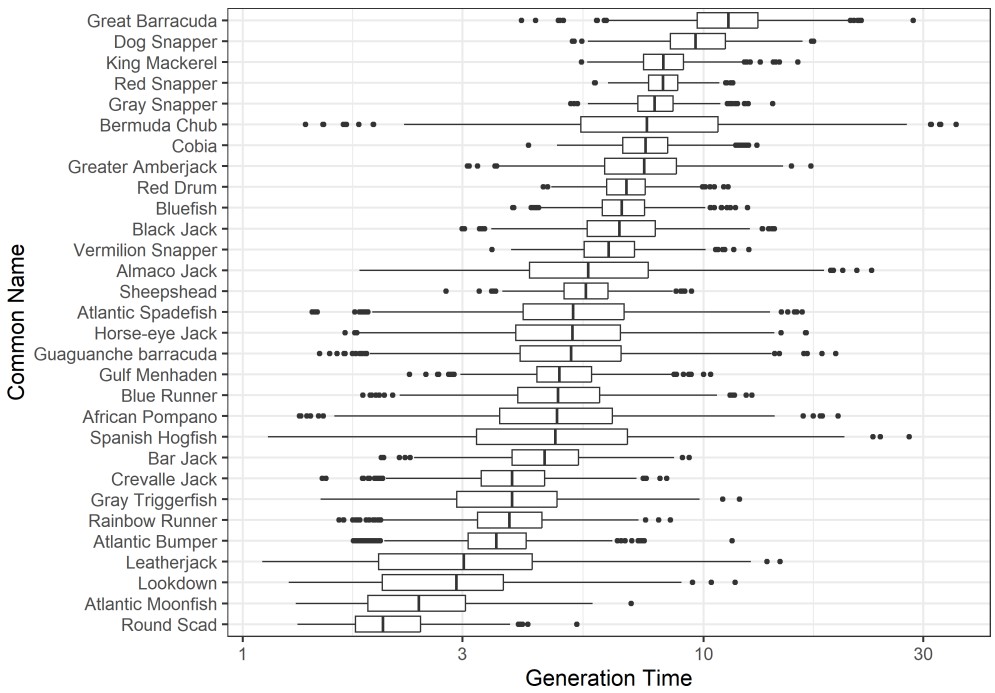

**Figure 2** **The generation time of 30 species of fish found at oil and gas platforms in the Northern Gulf of Mexico.** A box plot shows the median (vertical bar), upper and lower quartile (box), 1. 5 x Inter Quartile Range (IQR) below or above quartile range (whiskers), and outliers (dots).

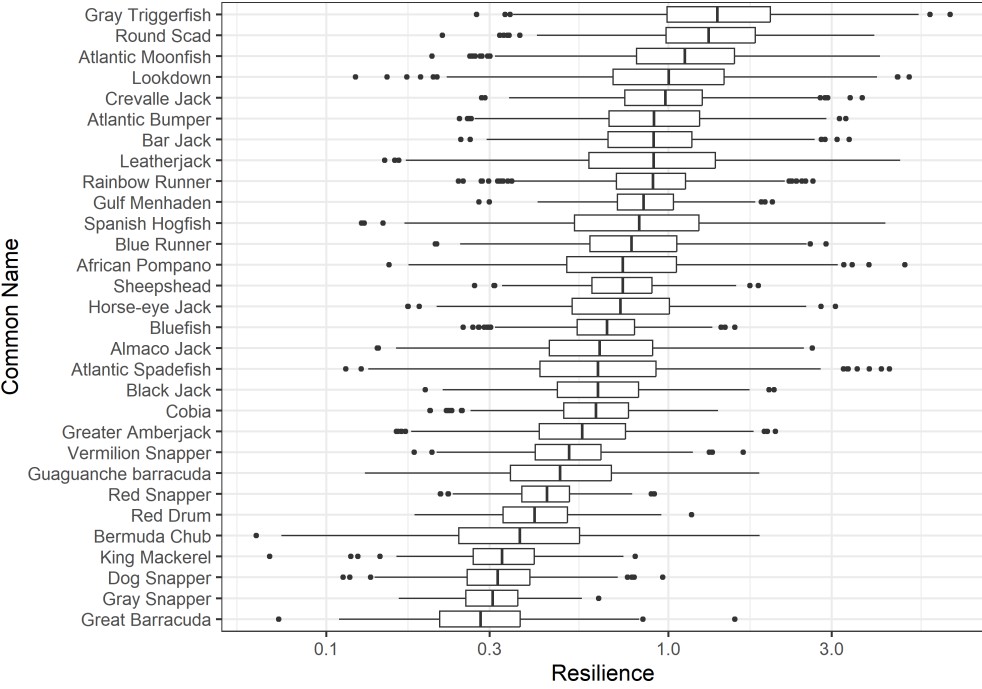

**Figure 3** **The resilience of the equilibrium point of 30 species of fish found at oil and gas platforms in the Northern Gulf of Mexico.** A box plot shows the median (vertical bar), upper and lower quartile (box), 1. 5 x Inter Quartile Range (IQR) below or above quartile range (whiskers), and outliers (dots).

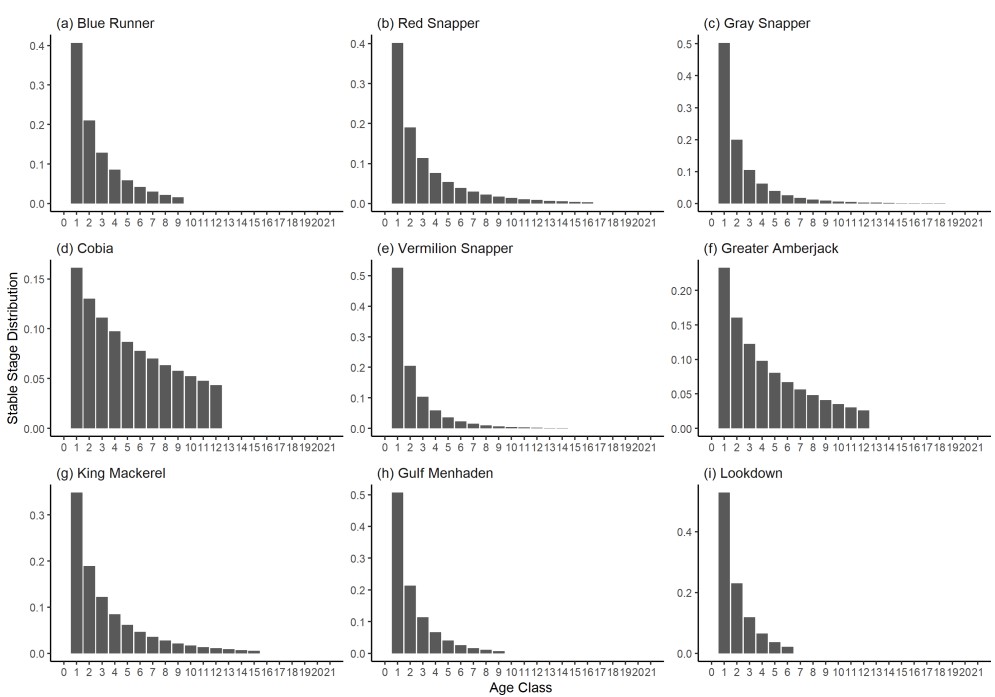

**Figure 4  The stable stage distributions of selected species of fish found at oil and gas platforms in the Northern Gulf of Mexico.** The bars show the proportion of individuals in each age class.

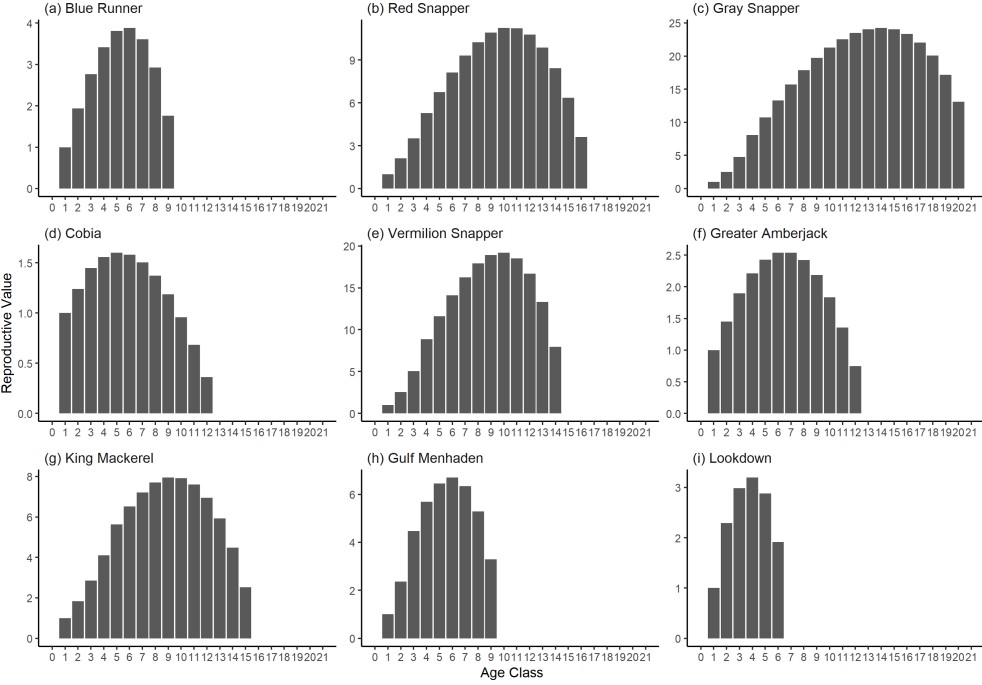

**Figure 5  The reproductive value of selected species of fish found at oil and gas platforms in the Northern Gulf of Mexico.** The bars show the relative reproductive value of individuals in each age class. The reproductive value was scaled so the first stage has the value of 1.

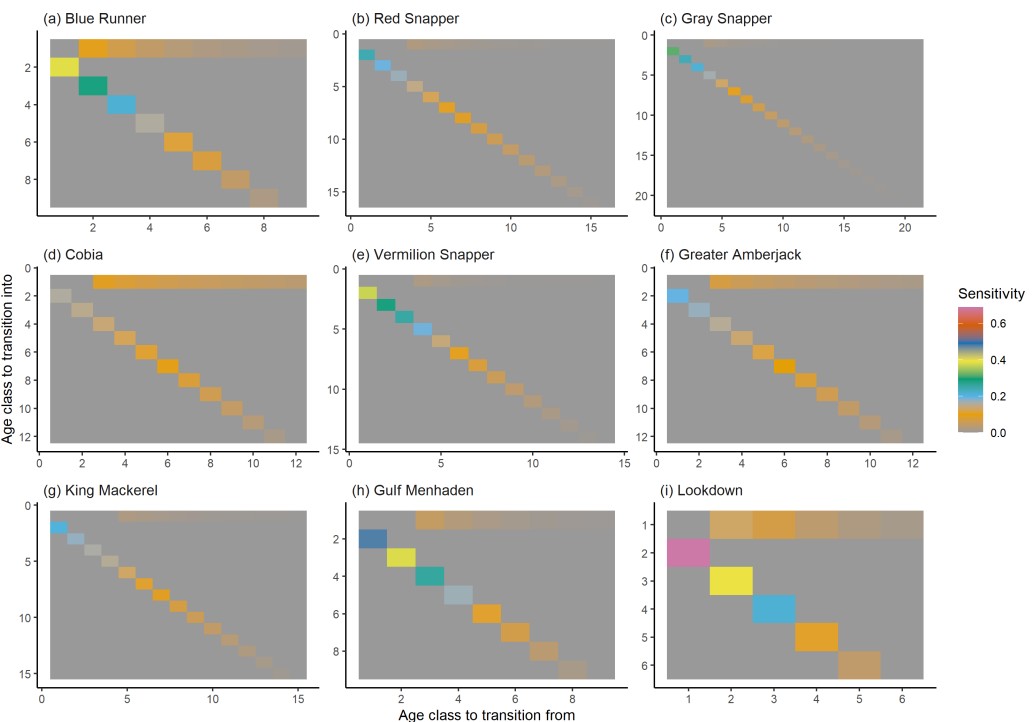

**Figure 6** The sensitivity of the asymptotic population growth rate to changes in the elements of a population matrix for selected species of fish found at oil and gas platforms in the Northern Gulf of Mexico. The color shows sensitivity. The horizontal axis corresponds to the column, and the vertical axis corresponds to the row of a matrix. Note the elements of the sensitivity matrix are set to 0 where corresponding elements of the population matrix are 0.

distribution, and reproductive value), can be used for comparisons among species. For example, *Heppell, Caswell & Crowder (2000)* propose the use of elasticity to categorize the life history strategies of mammals. Once species are categorized, similar management approaches may be applied to the species within the same category.

In addition to building population models for species with missing information, constructing standardized models across many species is often very useful in management because their metrics can be compared across all species (*e.g., Fujiwara, 2007*; *Fujiwara et al., 2011*; *Fujiwara, 2012*). The relationship between damping ratio and generation time highlights important ecological dynamics in species like greater barracuda and round scad. Specifically, longer generation times associated with lower damping ratios suggest that transient dynamics last longer for species like barracuda, requiring longer time series data to avoid misleading conclusions (*Wiedenmann, Fujiwara & Mangel, 2009*). The results also emphasize the importance of generation time in conservation efforts, as reflected in the IUCN Red List criteria (*IUCN, 2012*). Regarding resilience, species such as snapper may experience prolonged periods of low abundance even when fishing mortality is reduced, as seen in the slow recovery of red snapper populations (*NOAA Fisheries, 2023*). This underscores the importance of understanding resilience when setting recovery expectations and management goals. The patterns in stable age distribution and reproductive value

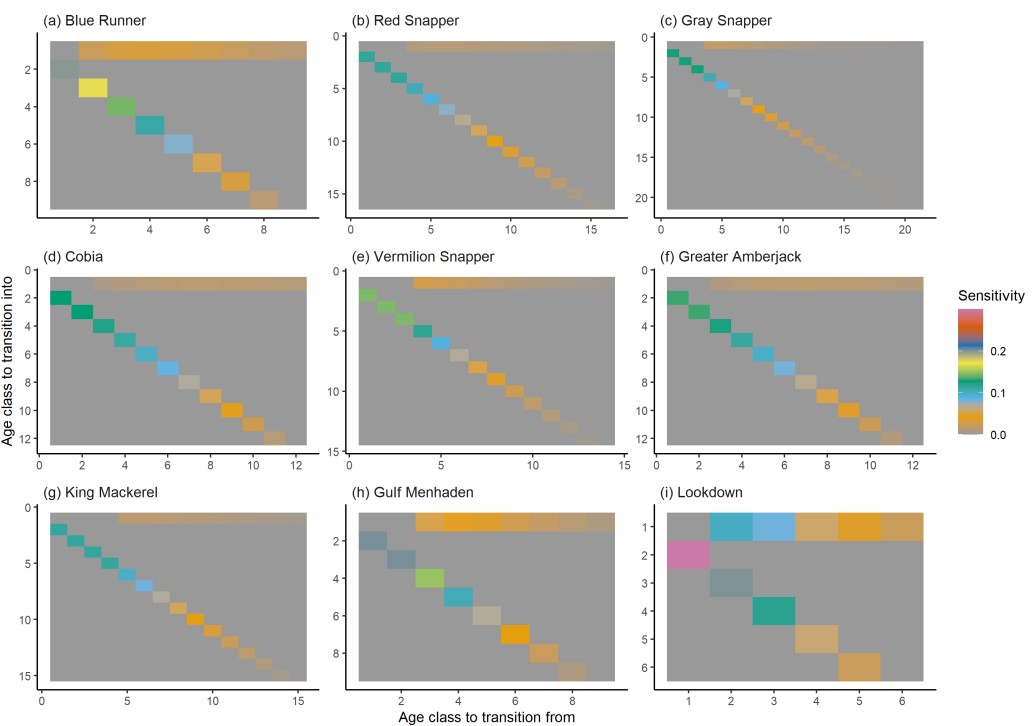

**Figure 7** **The elasticity of the asymptotic population growth rate to changes in the elements of a population matrix for selected species of fish found at oil and gas platforms in the Northern Gulf of Mexico.** The color shows elasticity. The horizontal axis corresponds to the column, and the vertical axis corresponds to the row of a matrix.

suggest different strategies for population growth across species. For example, cobia and greater amberjack rely more on younger individuals for future growth, while snapper species show a gradual increase in reproductive value with age, indicating the significance of older individuals in these populations. The sensitivity and elasticity matrices offer valuable insights for fisheries management. The relatively uniform elasticity values for cobia and greater amberjack suggest that management efforts should focus on reducing overall fishing mortality, while species like lookdown and gulf menhaden may benefit more from size limits that protect younger age classes.

The method presented in this paper relies on the estimates of life history parameters by the FishLife package (*Thorson, 2019*; *Thorson et al., 2017*). It assumes that taxonomic relatedness is a good predictor of life history parameters among species of fish. This assumption may need to be evaluated further. However, the purpose of the current study is not to evaluate the validation of Thorson's method. For example, *Heppell, Caswell & Crowder (2000)*, based on mammalian data, argue that life history parameters vary so much within a taxon compared with the overall variations. Future validations of the parameters predicted by the FishLife package as well as the matrix population models are probably needed. This may be done by evaluating prediction accuracy under different data-rich scenarios. The approach presented in this paper is a general method for integrating any

predictions of life history parameters, whether those methods are to be developed or already exist.

## CONCLUSIONS

This study demonstrates the method of parameterizing age-structured matrix population models using existing data, emphasizing the practical utility of these models. By leveraging extensive database resources and predictive algorithms, the algorithm represents a scalable approach applicable to a wide range of fish species, which can significantly enhance our understanding of fish population dynamics. This methodology supports more informed and effective conservation efforts, particularly for data-deficient species, and facilitates ecosystem-based management by enabling standardized metric comparisons across species, thereby contributing to the sustainability of marine ecosystems. The methodology comprises two main components: age-structured population models and life history parameters. Anticipated advancements in predicting life history parameters will further improve the utility of the method. The current models assume age as a reliable predictor of life history traits, which may not hold true for all organisms. Future research should focus on developing algorithms that parameterize population models based on developmental stages, size, or other relevant characteristics, to enhance model accuracy and applicability across diverse species.

## ACKNOWLEDGEMENTS

I would like to thank Taylor Beyea and Nathan Putman for providing the list of species found at the oil and gas platforms in the Northern Gulf of Mexico. I would also like to thank the anonymous reviewers and the editor for constructive comments that improved the previous version of this manuscript. The original manuscript was edited with a generative AI as described in the Methods.

### Funding

Funding was provided by the National Sea Grant Office in partnership with the U.S. Department of the Interior's Bureau of Safety and Environmental Enforcement (NA21OAR4170392). The funders had no role in study design, data collection and analysis, decision to publish, or preparation of the manuscript.

### Grant Disclosures

The following grant information was disclosed by the author:
The National Sea Grant Office in partnership with the U.S. Department of the Interior's Bureau of Safety and Environmental Enforcement: NA21OAR4170392.

### Competing Interests

The authors declare there are no competing interests.

## Author Contributions

- Masami Fujiwara conceived and designed the experiments, performed the experiments, analyzed the data, prepared figures and/or tables, authored or reviewed drafts of the article, and approved the final draft.

## Data Availability

The R scripts are available at Zenodo: Fujiwara M. 2024. R scripts for constructing age-structured matrix population models for all fishes. V1.0.1 ed: Zenodo. https://doi.org/10.5281/zenodo.11622238.

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
