# Peer review of "Constructing age-structured matrix population models for all fishes"

_PeerJ, doi:10.7717/peerj.18387_

## Round 0.1 · original submission · Major Revisions

It would be advantageous to provide a concise explanation of each of the seven key metrics mentioned in the abstract within the introduction. The author's effort to highlight the limitations of the models is commendable; however, incorporating a discussion of the limitations associated with utilizing an online database would enhance the depth of this paper. Addressing these limitations is crucial, as it acknowledges potential gaps or biases in the data and provides a more balanced perspective. The acknowledgment section is absent from the manuscript.

Reviewer 1 ·

Basic reporting

The reporting is clear and concise. The language is generally quite clear and readable. The analysis and methods had appropriate context given by the background; however, there is little context or depth of insight into the system represented by the case study. The article structure was quite unusual, with an extremely short results section, and descriptions of figures appearing in the discussion. While there were no clear hypotheses, it seemed as though some of the supporting material and motivation may be contained in an in press paper by the same author.

Experimental design

The mathematical structure of the framework is well described and the key output metrics are supported by prior demographic literature. While the methods were described in sufficient detail in the text, the code was not available for review and thus the function of the analysis could not be assessed. As noted, there is a need to fill in gaps in life history information to inform EBFM, and thus the work is relevant and meaningful in terms of a statistical methods development that builds on trait databases. However, a well-defined research question is lacking here. The rigor of the analysis was somewhat limited by a cursory review of specific results from a few of the 30 species analyzed.

Validity of the findings

The data and code were not available for review, but an in-text citation indicates that the author intends to put the code on Zenodo for access at a future time. The general conclusions are supported intuitively, in that this is a method for filling in data gaps to compute matrix models for all fishes by relying on these large trait databases. However, no specific questions were set up in the context of the case study analysis. Nor was there a broader characterization of the potential skill or error of the proposed method for evaluating the realism of these matrix model parameterizations based on public data and phylogenetic interpolation.

Additional comments

General comments:
This manuscript applies a flexible matrix population model framework to 30 fish populations by filling in missing life history information using public trait databases and phylogenetic inference tools. This approach does a great job of leveraging FishBase and FishLife infrastructure to create age-structured models for a much larger number of species within an assemblage than previously possible. The methods are sensible and the outputs are generally interpretable into their lessons for management. However, the case study that the framework was applied to was not thoroughly motivated or rigorously analyzed in a way that provides scientific value though revealing something about the system to which this is applied. The manuscript could be improved by providing more context for the species and system analyzed to show a thoroughly worked example of how the framework could be applied to address scientific questions. Alternatively, if there was a way to further the validation of the general approach, even through simulation, to show how accurate and precise the predictions can be given different data richness, that would also be a great contribution.

Specific comments:
Line 3: “all fishes” seems like an overstatement given that the demonstrated application is to only 30 species in one area.
Line 74: Reliability of fishbase information is questionable, just need to provide that caveat at least
Line 111-113: It would be interesting to see how the results differ when you make different assumptions about fecundity at age.
Lines 258-269: results too short and general, not including a full description of the figures. Instead, much of this goes into the discussion, which doesn't seem appropriate.
Lines 305-358: this should mostly go into results, then the discussion should build more on what these results tell us relative to some hypothesis, rather than simply explaining the general utility for management of these metrics and noting some post-hoc contrasts among species. In addition, it would be valuable to have a table that includes the actual life-history parameters for the reader to see how their life history influences these derived measures from the matrix models.

Reviewer 2 ·

Basic reporting

No comment

Experimental design

No comment

Validity of the findings

No comment

Additional comments

1. It would be great to explain succinctly each of the seven key metrics mentioned in the abstract in the introduction.
2. I commend the author for stating limitations of the models highlighted, however, including a discussion on the limitations of using an online database would enhance the depth of this paper. Addressing these limitations is important, as it would acknowledge potential gaps or biases in the data and provide a more balanced perspective.
3. Acknowledgment is missing
4. Figure 4 and 5 x-axis limits should stay consistent for easy and direct comparison over the different species.
5. There seems to be some inconsistencies with the two phrases- Stable Stage distribution and Stable Age distribution. Kindly look into it.

Annotated reviews are not available for download in order to protect the identity of reviewers who chose to remain anonymous.

---

## Round 0.2 · accepted · Accept

Many thanks for addressing all of the reviewers' comments. Based on the review of the present version, I believe the manuscript is ready for publication.

Reviewer 1 ·

Basic reporting

Good

Experimental design

Good

Validity of the findings

Good